# Host heterogeneity and epistasis explain punctuated evolution of SARS-CoV-2

**Bjarke Frost Nielsen**[1,2]*, **Chadi M. Saad-Roy**[3,4], **Yimei Li**[5], **Kim Sneppen**[2], **Lone Simonsen**[1], **Cécile Viboud**[6], **Simon A. Levin**[5], **Bryan T. Grenfell**[5]

**1** Department of Science and Environment, Roskilde University, Roskilde, Denmark, **2** Niels Bohr Institute, University of Copenhagen, Copenhagen, Denmark, **3** Department of Integrative Biology, University of California, Berkeley, California, United States of America, **4** Miller Institute for Basic Research in Science, University of California, Berkeley, California, United States of America, **5** Department of Ecology and Evolutionary Biology, Princeton University, Princeton, New Jersey, United States of America, **6** Division of International Epidemiology and Population Studies, Fogarty International Center, National Institutes of Health, Bethesda, Maryland, United States of America

* bjarkefrost@ruc.dk

**Data Availability Statement:** Data processing and model simulation code is available as a GitHub repository at https://github.com/BjarkeFN/Saltation.

## Abstract

Identifying drivers of viral diversity is key to understanding the evolutionary as well as epidemiological dynamics of the COVID-19 pandemic. Using rich viral genomic data sets, we show that periods of steadily rising diversity have been punctuated by sudden, enormous increases followed by similarly abrupt collapses of diversity. We introduce a mechanistic model of saltational evolution with epistasis and demonstrate that these features parsimoniously account for the observed temporal dynamics of inter-genomic diversity. Our results provide support for recent proposals that saltational evolution may be a signature feature of SARS-CoV-2, allowing the pathogen to more readily evolve highly transmissible variants. These findings lend theoretical support to a heightened awareness of biological contexts where increased diversification may occur. They also underline the power of pathogen genomics and other surveillance streams in clarifying the phylodynamics of emerging and endemic infections. In public health terms, our results further underline the importance of equitable distribution of up-to-date vaccines.

## Author summary

The coronavirus responsible for the COVID-19 pandemic, SARS-CoV-2, has shown a remarkable ability to evolve novel, increasingly transmissible variants. Using large amounts of viral sequences sampled during the pandemic, we map the genomic diversity over time. We find that the pathogen has followed a clear pattern of punctuated evolution, where periods of genetic drift are interrupted by sudden large increases in diversity followed by similarly abrupt collapses. This is in contrast to the pattern previously identified for influenza, which does not show similarly sudden increases in diversity. Using a mathematical model, we show that the observed pattern can result from rare evolutionary jumps (saltations) occurring within some hosts, in combination with epistasis. One possible explanation for such jumps is accelerated evolution within immunocompromised hosts,

**Funding:** BFN and LS received funding from the Carlsberg Foundation under its Semper Ardens programme (grant CF20-0046). BTG acknowledges financial support from the Flu Lab and the Schmidt DataX Fund at Princeton University made possible through a major gift from the Schmidt Futures Foundation. SAL acknowledges the support of the the C3.ai Digital Transformation Institute and Microsoft Corporation, Gift from Google and the National Science Foundation (CNS-2027908, CCF1917819). CMSR acknowledges funding from the Miller Institute for Basic Research in Science of UC Berkeley via a Miller Research Fellowship. YL was supported by a gift from William H. Miller III to SAL's research. KS received funding from the European Research Council (ERC) under the European Union's Horizon 2020 research and innovation program under Grant Agreement No. 740704. The funders had no role in study design, data collection and analysis, decision to publish, or preparation of the manuscript.

**Competing interests:** The authors declare no competing interests.

underscoring the importance of equitable vaccine distribution. Furthermore, a simple modification of the model to include incomplete cross immunity offers an explanation for recently observed patterns of variant co-circulation.

## Introduction

During the coronavirus disease 2019 (COVID-19) pandemic, the responsible pathogen, severe acute respiratory syndrome coronavirus 2 (SARS-CoV-2), has continuously evolved. However, evolution has by no means happened at an even pace, but rather through a pattern of steady diversification punctuated by sudden large jumps involving dozens of point mutations. Indeed, it has been suggested that SARS-CoV-2 exhibits saltational evolution, a process where evolution proceeds by large multimutational jumps, rather than gradually [1].

A simple way to quantify the genomic diversity existing at a given time is through the pairwise Hamming distance. Given two genomes, the pairwise Hamming distance simply measures how many nucleotides the two sequences disagree on. This rather crude measure turns out to reveal surprisingly robust patterns of viral diversification.

Due to the large amount of full genome sequencing performed on SARS-CoV-2 specimens during the COVID-19 pandemic, Hamming distances can be computed not just at the level of summary statistics, but as temporally varying distributions (Fig 1; S1 Video), revealing a pattern of slowly increasing diversity punctuated by abrupt increases and subsequent collapses in diversity.

That much can be gleaned from considering the time-development of the mean (or median) Hamming distance. However, the dynamics of the often multimodal distribution is not captured by the mean Hamming distance, even if temporally resolved, and much less by the usual static treatment. The full time-dependent Hamming distribution possesses further structure, which reveals that successive variants are well-separated in sequence space; this suggests that one did not arise from the other by a string of single-point mutations accruing in successive hosts. Rather, an evolutionary jump—a saltation—seems to have taken place at each major transition (see S1 Video). Recently, a somewhat different pattern of variant co-circulation and rapid turnover of variants has appeared—a phenomenon that we will also comment on in this paper, from the perspective of the Hamming distribution.

### Dynamical explanations

There are several plausible mechanisms that may contribute to saltational evolution in SARS-CoV-2, including increased build-up of mutations in immunocompromised individuals infected with SARS-CoV-2 [1, 3–9] and evolution in animal reservoirs followed by animal-to-human transmission [10, 11].

In this paper, we present a mathematical model aimed at capturing the particular punctuated evolutionary pattern of SARS-CoV-2. Our goal is to recapitulate the main features of the temporal Hamming distribution observed during the COVID-19 pandemic (see Fig 1A) as parsimoniously as possible in a dynamical model.

We show that the overall pattern can be captured by combining epistasis with heterogeneous within-host evolution. The model is sufficiently general that it does not make any assumptions about the detailed biological mechanism behind saltations.

The proposed model is conceptually related to the NK Model of [12] and [13] in that it operates on the space of possible genotypes, with each genotype corresponding to a preassigned fitness value. This is in contrast to phenotypic fitness landscape models which operate

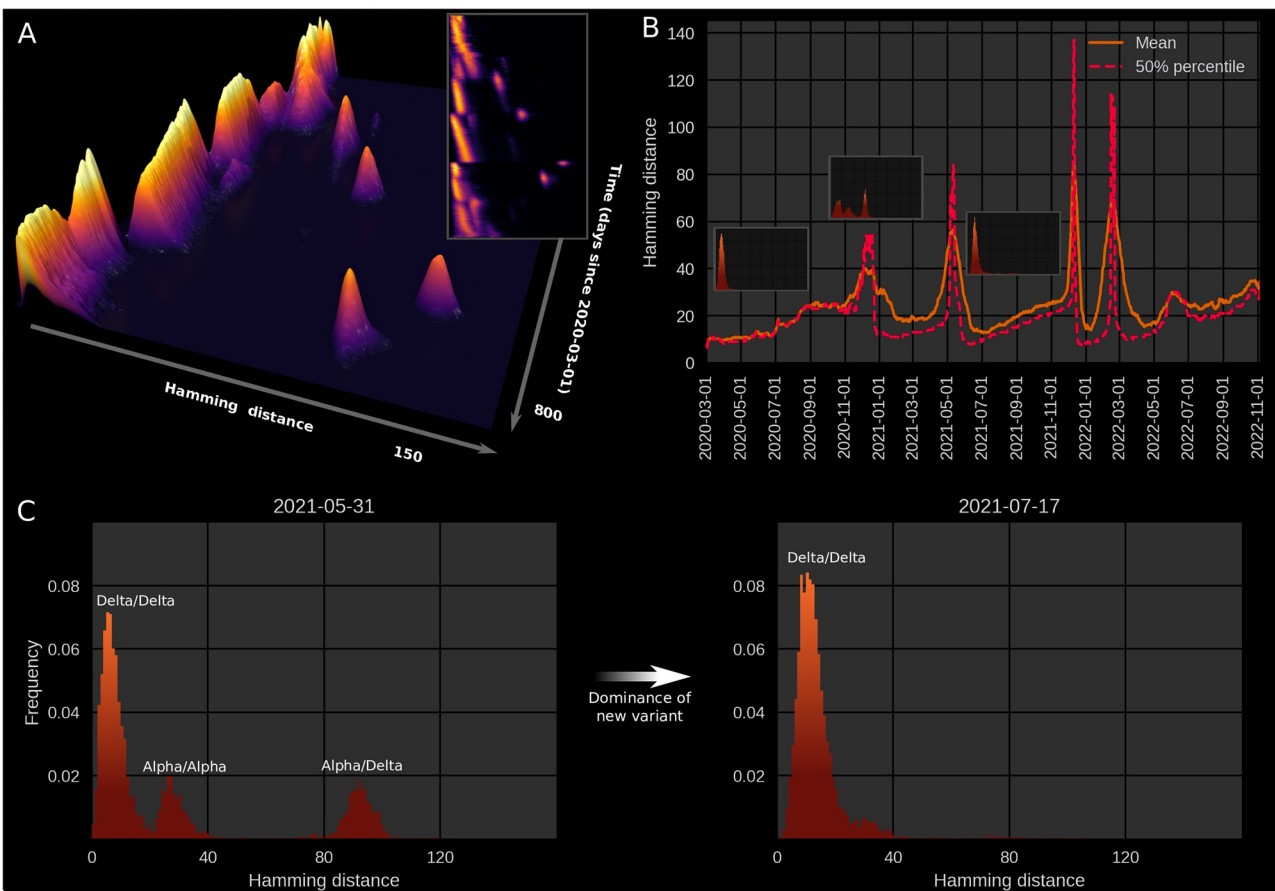

**Fig 1. Genomic diversity over time in SARS-CoV-2, UK genomic sureveillance data. (A)** Full, time-dependent Hamming distance distribution (UK data, GenBank via Nextstrain [2]). The 3D map shows the period 2020–03-01 to 2022–05-10, to focus on the major jumps. The insert shows a two-dimensional heatmap representation of the time-dependent Hamming distribution for the entire data range, 2020–03-01 to 2022–11-11. **(B)** Time evolution of the mean and median Hamming distance for the date range 2020–03-01 to 2022–11-11. Each time point represents Hamming distances between genomes sampled within a one-week window beginning on that date. The three miniature inserts show Hamming distance histograms at three different time points. **(C)** Left: A snapshot of the Hamming distance distribution for genomes sampled during a one-week window starting on May 31st, 2021. The three distinct peaks correspond to the Hamming distances between pairs of genomes from each of the prevailing variants at the time, Alpha and Delta. Right: 47 days later, a single variant (Delta) dominates. Related supporting figures: S1 and S2 Figs. See also S1 Video for the animated Hamming histogram.

directly on the space of possible values of some quantifiable trait. The most well-known among those is perhaps Fisher's geometric model [14] which assumes a continuous phenotypic ('trait') space with a single optimum [15] and that the effects of single mutations are mild [16]. The NK Model, a genotypic fitness landscape model, instead explicitly allows for a rough (epistatic) fitness landscape. The NK model, however, does not include the concept of *neutral space*—in that model, mutations are generically accompanied by a change in fitness. Our model includes neutral mutations and is, in that respect, closer to the models of [17, 18].

However, a crucial component of our model is the presence of sign epistasis, i.e. that the fitness contribution of a point mutation may change sign (going from deleterious to beneficial or vice-versa) depending on the presence of other mutations. This property turns out to offer an explanation for the role of saltation in evolving high-fitness SARS-CoV-2 genotypes.

In a recent study, Starr et al. [19] showed by deep mutational scanning that epistasis—including sign epistasis—is an important feature of SARS-CoV-2 evolution. As a concrete example, they show that the N501Y mutation (which is present in the Alpha, Beta and

Omicron variants) and Q498R exhibit sign epistasis. In this case, the presence of the N501Y substitution changes the contribution of Q498R from deleterious to advantageous, as measured by angiotensin-converting enzyme 2 (ACE2) receptor binding affinity.

In general, the fitness landscape of an organism is combinatorially large, and the number of possible evolutionary paths from one genotype to another fitter one is, a priori, enormous. However, in seminal works, Weinreich et al. [20, 21] showed that only very few such paths are in fact accessible. The interpretation of this finding in terms of fitness landscapes is that epistasis or the *ruggedness* of the landscape is highly important for understanding evolutionary trajectories [15]. However, even if evolutionary paths seem blocked, this conclusion may only hold in the weak mutation limit, i.e. when the probability of multiple mutations arising in the same genome within a generation is low [22]. If saltational evolution is possible, even seemingly inaccessible regions of the fitness landscape may be explored by the organism. Our model suggests that such saltations may thus increase—or, in some cases, altogether enable— the emergence of new concerning variants.

## Results

### SARS-CoV-2 genomic diversity is characterized by punctuated evolution

On the basis of UK sequences (a particularly rich data set), we have computed a time-dependent Hamming distribution for SARS-CoV-2, which is presented in Fig 1. Fig 1A shows the full Hamming distance histogram as a function of time, from March 2020 to mid-2022, with the colour and height indicating the frequency of observing sequence pairs with a particular Hamming distance. The peaks that correspond to saltational variant transitions are clearly visible as isolated 'islands' at large Hamming distance. The insert in the same panel shows a 2D heatmap representation of the data, including data up to mid-November 2022.

In panel B, time series of the mean and median Hamming distances are shown, revealing clear spikes associated with each of the major variant transitions, ancestral variant→Alpha, Alpha→Delta, Delta→Omicron (BA.1) as well as Omicron BA.1→BA.2 (by "ancestral variant", we mean the lineages circulating before the Alpha transition, whether including the D614G substitution or not [23, 24]). Each of these transition events is marked by a very sudden spike in the typical Hamming distance, as is especially clear when considering the median (Fig 1B, dashed line) which increases almost discontinuously at these transitions. It should be noted that data quality is highest after the end of 2020, when sequencing capacity was greatly increased, and before February 2022. As a concrete example, 4,945 sequences were included for June of 2020, while 72,292 sequences were included for the month of June, 2021.

In Fig 1C (left), a snapshot from May 31st 2021 shows three well-defined peaks. Each peak corresponds to comparisons between pairs of genomes, with the members of each pair belonging to either the Alpha or Delta variant. The peak corresponding to the highest Hamming distance is of course that due to comparisons between the 'new' and 'old' variant, since these are furthest from each other in a genomic sense. Similarly, the plot clearly shows that variation within the Delta variant is, at that point in time, much lower than within the Alpha variant, since each of the Delta variant genomes belong to a clade with a recent common ancestor. In the right half of Fig 1C, the situation 47 days later is shown, once the Hamming distribution has collapsed to a single peak, corresponding to the then-dominant Delta variant.

During the month of March, 2020, the Hamming distribution appears bimodal, but there are no signs of saltation. This transient bimodality, present in the early pandemic, can most clearly be seen in S1 Video. This can be explained by the D614G substitution, which was associated with a clade that dominated from around the end of March/beginning of April 2020 [25]. This early, saltation-free transition is reminiscent of a result by [12], who suggested that

adaptation on a rugged fitness landscape is associated with two separate time scales. First, the pathogen searches its neighbourhood in the fitness landscape until it finds a local maximum. This does not require saltation and happens rather rapidly. Then, on a slower time scale, the pathogen may transition to new fitness peaks by saltation.

Due to the relatively high quality of SARS-CoV-2 genomic surveillance in the United Kingdom, both in terms of the absolute number of publicly available sequences and per capita coverage, we have based the bulk of our observations on UK sequence data. However, patterns similar to those presented here can be observed in US data, the analysis of which is included in S1 Appendix.

For each day in the included range (2020–03-01 to 2022–05-10) a 7-day window (consisting of the indicated day and the 6 following days) was considered. All high-quality sequences obtained within that window were pooled, and a distribution of Hamming distances was compiled by repeatedly picking out random pairs from the sequence pool and comparing.

While the Hamming distance is a somewhat crude measure of the variance between circulating genomes, it turns out to offer a surprisingly powerful window into the evolution of SARS-CoV-2 when large amounts of sequence data are available. The aforementioned transitions all show the tell-tale signs of saltational evolution, i.e. sharply increasing typical Hamming distances which appear as clearly defined, disconnected 'islands' in the full distribution (Fig 1A). The Omicron BA.2→BA.5 transition is less clearly defined, although a moderately sized genetic jump does appear to be present in the data. It should be noted that UK sequencing has become less dense since February 2022, meaning that there is not as much data for the BA.2→BA.5 transition. The BA.2→BA.5 transition was also muddled somewhat by the BA.2.12.1 subvariant briefly making up as much as 10% of UK sequences [26]. The main part of our analysis is focused on the four saltational transitions mentioned above. Since the appearance of the Omicron subvariant BA.5, the simple picture of periods of linearly increasing Hamming distance interrupted by saltations has been replaced by a higher degree of variant coexistence and rapid turnover. We comment on this recent situation and how it may fit into our modeling framework in S3 Appendix as well as in the Discussion.

As shown in S2 Appendix, all but one of the saltational transitions are also associated with a discontinuous increase in the distance to the *origin* (Wuhan-Hu-1, GenBank reference sequence accession number MN908947.3). The exception is the Alpha→Delta transition, where a moderate decrease is observed. In other words, the Delta variant is closer to the ancestral variant than Alpha is. In S2 Appendix, we model one possible explanation for this phenomenon, namely the occurrence of persistent infections.

The plots of Fig 1 are based on the entire SARS-CoV-2 genome, meaning that a substitution leading to an amino acid change in the spike protein (a major antigen) counts just as much as a synonymous mutation elsewhere in the genome. In Fig 2, we probe to what extent the observed drift-boom-bust pattern of diversity is driven by changes in the S-gene (coding for the spike protein) or by (non-)synonymous mutations. Overall, the pattern is present whether considering only the S-gene (Fig 2B), non-synonymous mutations (Fig 2C) or the entire genome (Fig 2A). We interpret this to mean that

1. The drift seen between saltations is not driven solely by synonymous mutations but affects the amino acid sequence as well.

2. When saltations occur, mutations are observed within the spike protein as well as outside it.

3. The observed pattern is quite robust, being observed within the whole genome, in the amino acid sequence as well as within the S-gene itself.

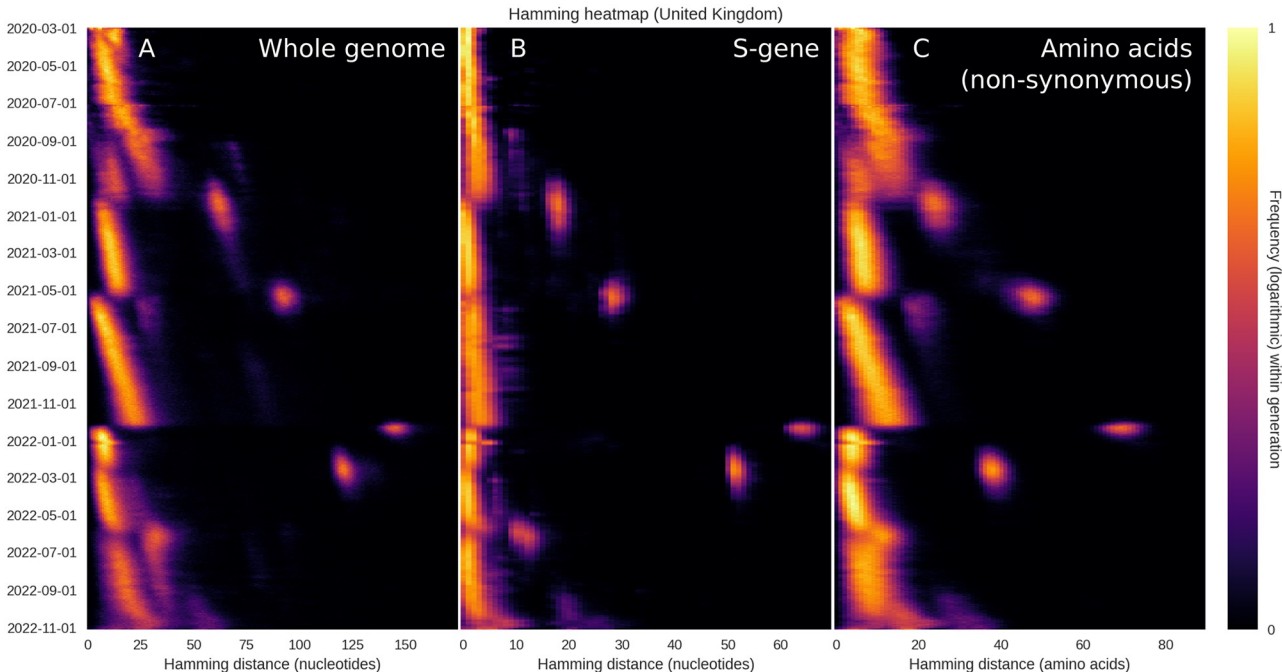

**Fig 2. Restricting the Hamming distribution to the S-gene or the amino acid sequence.** The overall temporal pattern of diversity seen in Fig 1 is found to persist when the analysis is restricted to the S-gene or non-synonymous mutations. **(A)** Temporal Hamming distance distribution based on the whole genome, included for reference. For each time on the vertical axis, the colour encodes the histogram of Hamming distances between genomes sampled within a one-week window starting on that date. **(B)** Temporal Hamming distance distribution confined to the S-gene which encodes the SARS-CoV-2 Spike protein. **(C)** Time evolution of the Hamming distance distribution as measured by the number of amino acid changes. We use this as a proxy for non-synonymous mutations, since a synonymous mutation would not produce an amino acid change.

It is notable, however, that the S-gene does not undergo quite as much drift as the whole genome, relatively speaking. That is to say, when the whole genome is considered, the Delta→Omicron jump is associated with a peak that is approximately 5.5 times larger than the typical Hamming distances in the weeks that preceded it, while the ratio is closer to 11 for the spike protein. We interpret this to mean that, while the S-gene is subject to large saltations, it undergoes less drift than an average, similarly sized section of the genome.

## Mechanistic modelling captures the essential dynamics

Our goal is to capture the overall temporal pattern of diversity observed in Fig 1 in a mathematical model that is as parsimonious as possible. The model consists of two parts: a branching process and an evolutionary algorithm incorporating sign epistasis and saltational evolution. Details of both elements can be found in the Materials and methods section. See also S3 Fig for a schematic description of the model elements.

The model assumes the existence of a number of possible high-fitness genotypes, but that each of them are 'screened' by epistasis. From a fitness landscape viewpoint, this can be thought of as a landscape with a number of peaks, each of which is surrounded by a fitness trough or valley. The extent of sign epistasis is then determined by the depth (and width) of these valleys.

To get from a moderate-fitness genotype to a local fitness peak, it is thus necessary to either traverse a region of low fitness, with its potential for extinction, or to somehow jump across that valley.

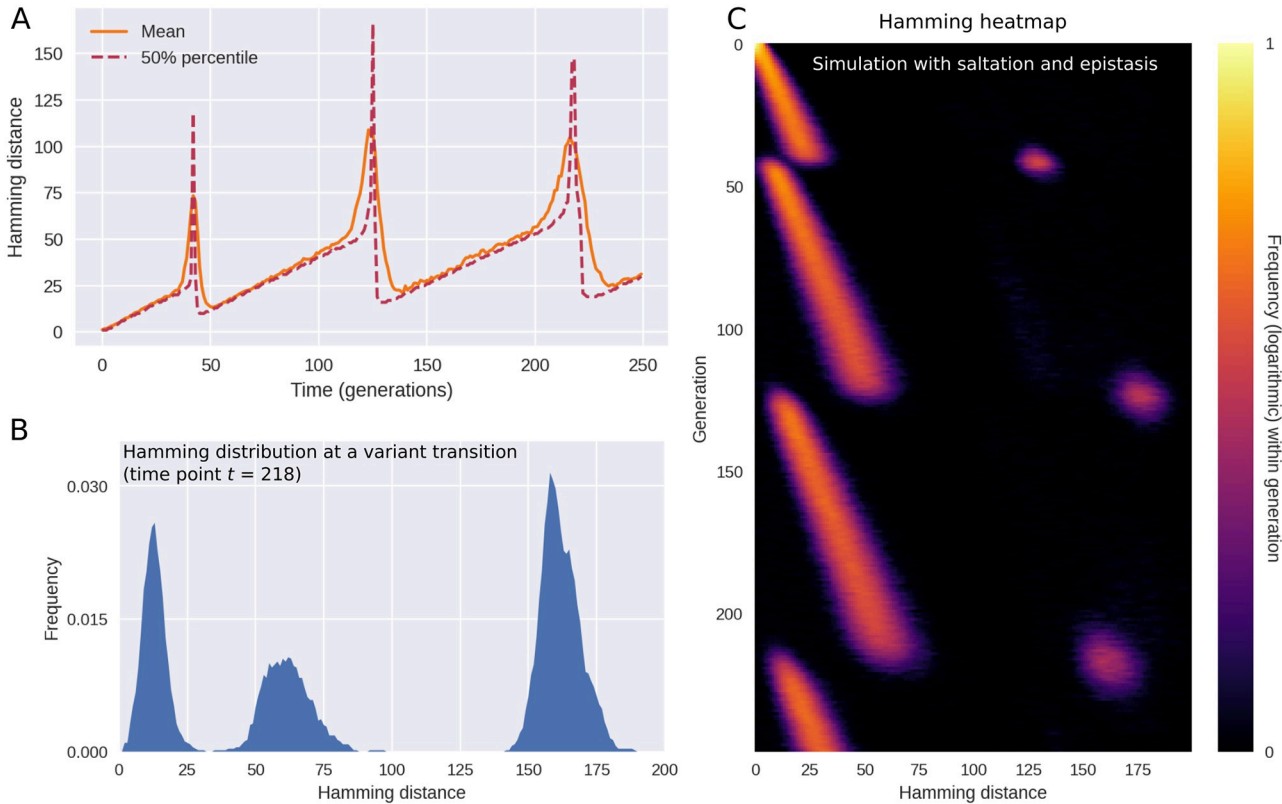

**Fig 3. Simulated outbreak with saltation (heterogeneous mutation rates) and epistasis. (A)** Time evolution of the mean and median Hamming distance between bitstring genomes present in any given generation of the model simulation. The pattern of genetic drift punctuated by sudden increases and subsequent collapses in diversity is similar to what is observed in SARS-CoV-2 (see Fig 1). **(B)** A snapshot of the Hamming distance distribution in generation $t = 218$ of the simulated outbreak. Just as in Fig 1, the three distinct peaks correspond to the distances between pairs of genomes from each of the two prevailing variants at the time. **(C)** Time evolution of the full Hamming distance distribution. For each generation on the vertical axis, the colour encodes the histogram of Hamming distances between genomes within that generation. The parameters used in these simulations were $\epsilon = 0.0001$, $d_0 = 3$, $\delta R_H = 1.0$, $\delta R_L = -\infty$ (i.e. deleterious mutations were fatal to the pathogen). Related supporting figures: S3 and S4 Figs.

Evolutionary models typically assume that the 'weak mutation limit' holds, meaning that the probability of several mutations arising in one genome in one generation is negligible [22], leading to gradual evolution. However, as described in the introduction, there are several mechanisms which can introduce a sudden burst of novelty within a single host, including by recombination [27–29]. The most well-documented is perhaps elevated mutation in immuno-compromised individuals [4, 30–32]. Our model, however, is agnostic with respect to the precise etiology, but includes saltation simply as rare occurrences of drastically increased evolution within a single host.

As shown in Fig 3, the model replicates the main features observed in Fig 1, including the long periods of drift (linearly increasing pairwise Hamming distances) punctuated by rapid rises and subsequent collapses of diversity. Just as in the empirical data, each variant transition is accompanied by three distinct peaks in the Hamming distribution.

The pattern shown in Fig 3 is the typical outcome of a model simulation, but occasional coexistence of two variants does occur in the model, see S4 Fig. This happens when two distinct variants with the same fitness happen to arise close to each other in time.

In the interest of simplicity, we have assumed an epidemic of constant size (constant incidence), however we explore the consequences of relaxing this assumption in the section

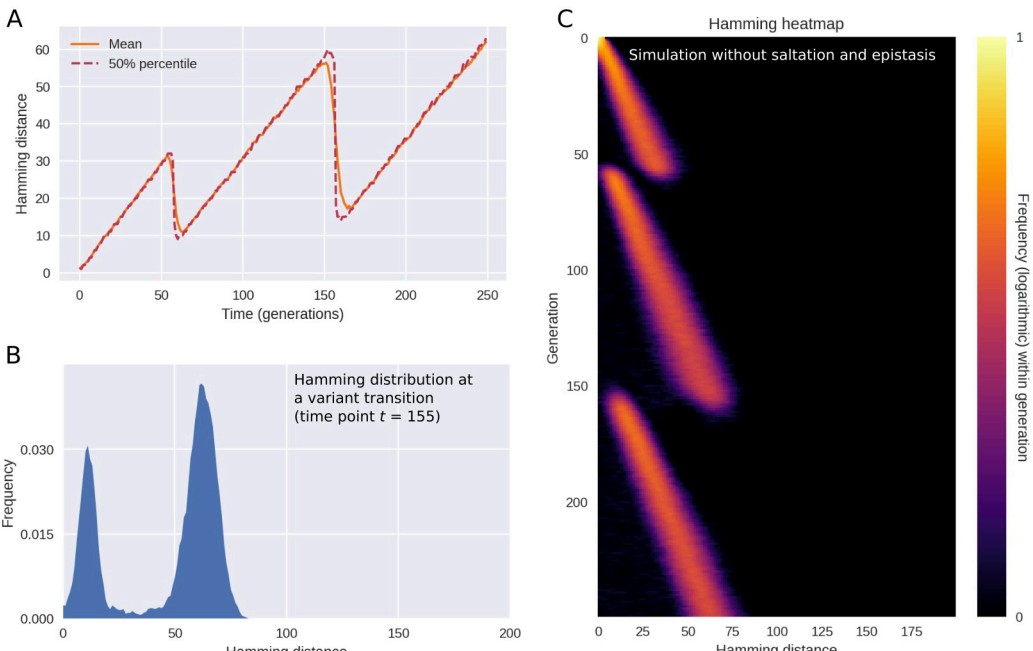

**Fig 4. In the absence of epistasis and saltation, model results do not match observations.** In these simulations, saltations do not occur ($\epsilon = 0$) and sign epistasis is absent ($\delta R_L = 0$). When a new pathogen variant emerges, the transition is marked by a collapse of diversity (as measured by the typical Hamming distance), giving a drift-bust-drift dynamics as opposed to the drift-boom-bust pattern seen in SARS-CoV-2. **(A)** Time evolution of the mean and median Hamming distance between genomes present in any given generation of the model simulation. **(B)** A snapshot of the Hamming distance distribution for bitstring genomes at generation $t = 112$ of the simulated outbreak. **(C)** Time evolution of the Hamming distance distribution. For each generation indicated on the vertical axis, the colour encodes the histogram of Hamming distances between genomes within that generation. Related supporting figure: S5 Fig.

*Epidemic dynamics and spatial structure.* Note that time in the model is measured (discretely) in generations, meaning that constant incidence and prevalence both hold.

If epistasis and saltation are turned off, evolution and variant transitions still happen within the model. The temporal pattern changes, however. In Fig 4, we explore this regime by setting $\epsilon$, the frequency of saltations, to zero and letting $\delta R_L = 0$, thus disabling sign epistasis. The resulting behaviour is characterized by periods of increasing diversity—essentially, genetic drift—interrupted by sudden collapses of the typical Hamming distance. No sudden spikes are seen in Fig 4A, rendering the dynamics fundamentally different from that of Figs 1 and 3. The behaviour observed in this regime is more reminiscent of the dynamics observed for H3N2 influenza in [17]. However, one could object that the temporal resolution of the empirical time series shown in [17] is not sufficiently high to allow one to discriminate between the scenarios of our Figs 3 and 4—after all, the periods of drastically increased pairwise nucleotide Hamming distance seen in Fig 1 are brief and require high temporal resolution to discern. While the amount of genomic data available for SARS-CoV-2 enables this, the picture is murkier for seasonal influenza. In S2 Fig, we present the result of applying the analysis of Fig 1A to influenza types H3N2 and H1N1. While there is no apparent evidence of saltation, the available data is relatively coarse-grained.

Another influential evolutionary model of influenza is due to [33]. In their model, the appearance of new variants is driven by immune system memory and a non-linear relation between Hamming distance and cross-immunity, the latter in the form of short-lived strain-

transcending immunity. While a sensible model for seasonal influenza, it gives rise to diversity dynamics which are closer to Fig 4 than to the pattern observed for SARS-CoV-2.

In the simulations of Fig 4, saltation and epistasis are completely lacking, but in S5 Fig, we consider what happens if some saltational evolution *does* occur, without sign epistasis. Qualitatively, the picture most resembles the saltation-free scenario of Fig 4, but occasional Hamming spikes are observed. Overall, this scenario does not conform to the empirical observations in the form of Fig 1. In the next section, we systematically probe how different levels of epistasis and saltation affects the evolution of new, highly transmissible variants.

Our focus is mainly on the dynamics of diversity, and for this reason we have emphasized the distribution of Hamming distances between viral genomes present in the population at the same time. This goes for the empirical observations (Fig 1) as well as our model simulations (Fig 3). However, in S2 Appendix, we explore the distributions of Hamming distance relative to the origin (meaning Wuhan-Hu-1, GenBank reference sequence accession number MN908947.3).

## Saltation facilitates the evolution of highly transmissible variants

Saltational evolution may not only be a way to generate vastly different variants, but may indeed be necessary for the virus to evolve highly fit variants at all. In the presence of strong epistasis, gradual evolution towards a high fitness genotype can be blocked (see S3A Fig). Conceptually, such gradual evolution under strong epistasis would correspond to traversing a deep valley in the fitness landscape by a series of small steps before reaching a peak [22, 34]. However, such a fitness valley indicates the presence of deleterious mutations which impart a high probability of extinction of the lineage in question, preventing the fitness peak from being reached.

In Fig 5, we explore how the strength of epistasis and the size of saltations affect the ability of the pathogen to evolve new, highly transmissible strains. By the *strength* of epistasis, we mean the typical depth of a valley in the fitness landscape, $|\delta R_L|$, i.e. the loss in reproductive number suffered.

For a pathogen which does not undergo saltational evolution (Fig 5A, dashed curve), significant sign epistasis ($|\delta R_L| \gtrsim 0.25$ at $d_0 = 3$ in our simulations) is a roadblock to evolution of

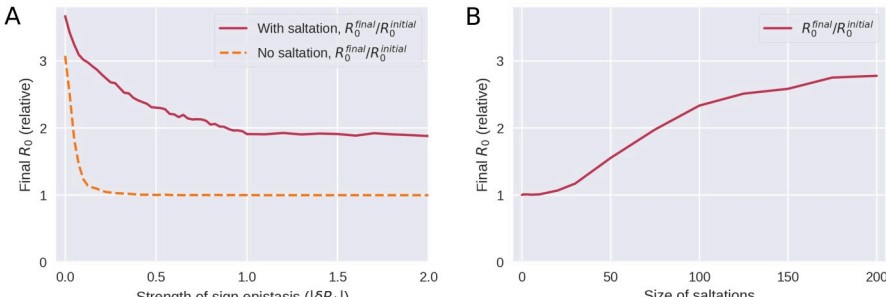

**Fig 5. Saltation allows highly transmissible variants to evolve by facilitating evolution across fitness landscape troughs. A)** Evolution under varying degrees of sign epistasis. The vertical axis indicates the final average reproductive number in the model population after 300 generations of the simulation, relative to (divided by) the reproductive number of the initial variant. The horizontal axis indicates the depth of a valley in the fitness landscape, $|\delta R_L|$, understood as the reduction in reproductive number suffered due to a deleterious configuration. Here, $\delta R_L$ was distributed according to a Dirac $\delta$ distribution and as such its value was deterministic. This panel is based on 90,000 simulations and the parameters used were $d_0 = 3$ and $\delta R_H = 1$. **B)** Evolution with varying degrees of saltation. Moderate sign epistasis is assumed ($\delta R_L = -0.5$). All other parameters are as in panel A. This panel is based on 7600 simulations.

high-fitness variants. However, a pathogen which undergoes saltation (fully drawn curve) can overcome this epistatic hindrance. Above a certain threshold (at $|\delta R_L| \approx 1$ in Fig 5A), stronger sign epistasis ceases to further impede the emergence of high-fitness variants. The mechanism behind this is that sign epistasis becomes so strong that a fitness valley may be overcome only by pure saltation and is no longer traversable by gradual evolution or a combination of the two.

As shown in Fig 5B, large saltations are necessary to overcome even moderate sign epistasis, further explaining why the Hamming peaks seen in SARS-CoV-2 are so large.

## Epidemic dynamics and spatial structure

In Fig 3, we made a number of simplifying assumptions, the major ones being constant prevalence and absence of any spatial or population structure. We first relax the former assumption by implementing susceptible-infected-recovered-susceptible (SIRS) dynamics. The infected individuals are now assumed to make up only a fraction of a larger population of total size $N$. Our aim is to ascertain whether the diversity dynamics observed in the previous section are fundamentally altered by allowing a variable number of infected individuals, $I(t)$, as well as susceptible depletion and waning immunity.

In Fig 6, a typical course of a simulation with SIRS dynamics, epistasis and saltation is shown. As shown in panel A, the number of recovered (immune) individuals varies non-monotonically over time, reflecting that individuals acquire immunity after being infected, and that the immunity eventually wanes. However, as successive variants of greater fitness

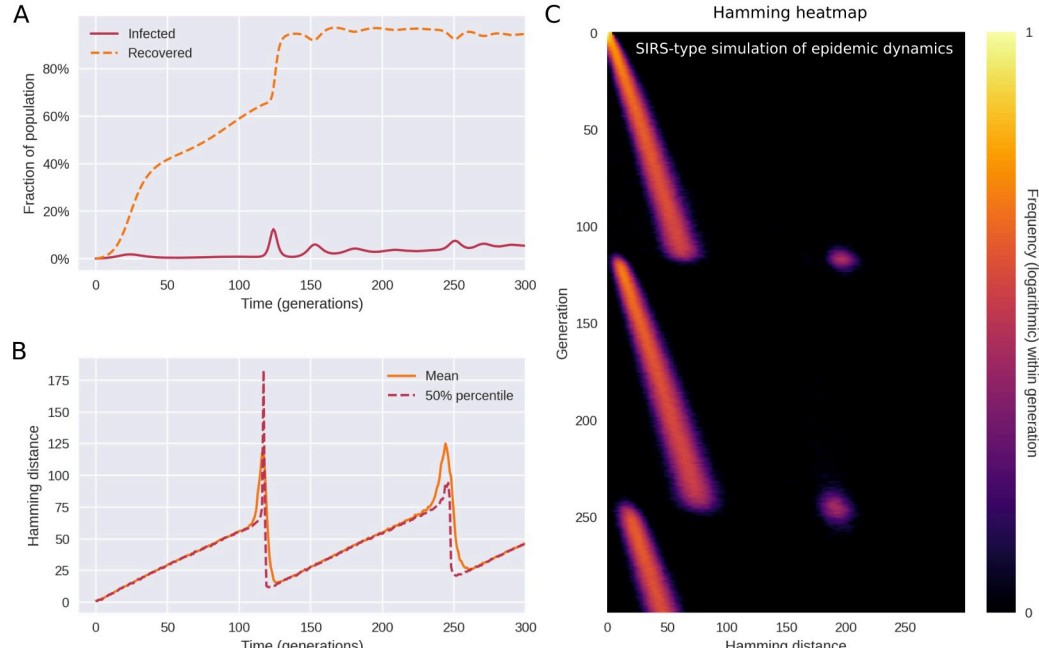

**Fig 6. Saltational evolution under susceptible-infected-recovered-susceptible (SIRS) dynamics.** The reproductive number of the initial variant is $R_0 = 1.2$ and immunity wanes at a rate of $\omega = 1/25$. Population size is $N = 2 \times 10^6$. The parameters of fitness-altering mutations are $\delta R_H = 1$ and $\delta R_L = -\infty$ (i.e. deleterious mutations are fatal to the organism or prevent transmission). **(A)** Time evolution of the recovered (or immune) fraction of the population. Successive variants have higher reproductive numbers ($R_0$), eventually leading to an endemic plateau (until further variants emerge). **(B)** Even under variable prevalence, the Hamming dynamics looks similar to that of Fig 3. **(C)** The full temporal Hamming distribution is characterized by the same kind of punctuated evolution as in the simpler constant-prevalence case of Fig 3. Related supporting figure: S6 Fig.

(greater reproductive number $R_0$) arise, an endemic plateau is eventually reached. While the epidemiology is very different from that of Fig 3, the Hamming distribution (Fig 6C) is remarkably similar. This indicates that the mechanism of saltational evolution in conjunction with sign epistasis robustly reproduces the punctuated evolutionary dynamics seen in Fig 1.

In simulations with variable incidence, higher incidence translates to an increased risk of emergence of new variants, all else being equal. Since saltations are simulated as a constant-rate (Poisson) process for each infected individual, the risk of emergence scales with the number of infected. Since simulations are stochastic, this tendency is not necessarily clear from a single realization such as Fig 6. A similar frequency dependence is likely to hold for SARS-CoV-2, since rare occurrences in terms of within-host evolution are proportionally more likely to be observed with higher incidence.

Next, we probe the impact of spatial separation on the diversity dynamics. Spatial structure is implemented by augmenting the model with a metapopulation element, see Materials and methods for details. We find that, if transmission between populations is limited (i.e. spatial effects are strong), variant transitions become protracted such that the transient multimodality of the Hamming distribution lasts longer. The duration of coexistence of strains with different fitness levels is observed to be determined by the transmission rate $\beta_{ij}$ between different populations (i.e. with $i \neq j$). In S6A Fig, we probe three situations where (relative) inter-population transmission rates are either 0, $10^{-4}$ or $10^{-3}$ (with intra-population transmission rates $T_{ii} \approx 1$). We find that with very low ($< 10^{-3}$) transmission rates between populations, spatial structure leads to drawn-out transitions, but that this effect disappears as soon as significant transmission between populations occur. This intuitively makes sense, since the within-population transmission rate will dominate as soon as just a few cases of a new variant have spilled into a population.

The addition of spatial structure also allows us to probe a potential source of apparent saltations. What if a new variant arises by gradual evolution within an unobserved—that is, unsequenced—population? Will an eventual spillover to the sequenced popoulation then give rise to an apparent saltational signal in the Hamming distribution, even if no actual saltations occur? To investigate this question in a simulation, we consider two populations which are not initially in contact with one another, see S6B Fig. We think of population 1 as the unobserved population (although the Hamming distances for this population are still given in the leftmost panel of S6B Fig). At time $t = 30$, we let a fitter variant arise in population 1 by only a few point mutations. This of course leads to a rapid decrease in the typical Hamming distance within population 1. At time $t = 70$, the two populations are then put into contact with each other (the relative inter-population transmission rate is increased from 0 to 0.5). This leads to a spillover of the fitter variant from population 1 into population 2. However, no apparent saltation results.

We conclude that spatial structure—including isolated populations—cannot by itself lead to the saltational signature seen in Fig 1. By this we mean that a pathogen which evolves gradually (i.e. obeys the weak mutation limit) in multiple spatial patches will not lead to a sudden spike in the Hamming distance once spillover happens. The reason for this is that two spatially remote lineages diverge from each other (in terms of Hamming distance) at approximately the same rate as less geographically distant pairs of lineages—they follow the same molecular clock.

In S7 Fig, we show Hamming distribution based on A) global sequences and B) sequences obtained outside of North America and Europe. First, it is worth noting that the distribution based on worldwide sequences as well as the plot based on sequences outside of Europa and North America give Hamming distances which are similar in magnitude to those obtained from e.g. just the United Kingdom. However, the number of sequences available outside of Europe

and North America is very low, so we cannot obtain a plot of similar quality. The main observation from S7B Fig is that transitions are more "smeared out" and that a higher degree of coexistence is observed. This is consistent with what we observed regarding spatial effects in S6 Fig.

## Discussion

The pattern of evolution observed in SARS-CoV-2 suggests that transmissibility of the pathogen has mainly increased due to large evolutionary 'jumps', rather than due to gradual evolution, something that may turn out to be a signature feature of the pathogen. Our model simulations highlight how this preference for adaptation by saltation may be explained by an ability to overcome epistatic 'fitness valleys'. The implications for public health are clear; any situation which facilitates such jumps should be treated with heightened awareness. They represent a high risk for the emergence of new, concerning variants which could not have emerged through gradual evolution. Below, we discuss and critique the implications of our results, as well as laying out directions for future work.

### Multiple possible sources of saltations

While much attention has rightly been given to the role of immunocompromised individuals, it is important to realize that other probable mechanisms of saltation exist. For instance, consider reverse zoonosis—the transmission from humans to animals. The epistatic landscape may be very different in animals, affording a way of bypassing what would otherwise be troughs in the human SARS-CoV-2 fitness landscape. Reintroduction of the mutated lineage into the human population would then constitute a 'jump' in terms of Hamming distance, and potentially also phenotypically. An example of such back-and-forth transmission between human and animal hosts leading to a large number of novel mutations was the so-called Cluster 5 variant, which evolved in mink (*Neovison vison*) in Denmark and subsequently spread to humans [35]. This mink-derived variant, which was only one of several which escaped into the human population, exhibited 35 substitutions and four deletions in the spike protein alone [11]. However, there is at present no strong evidence that reverse zoonosis explains the observed jumps associated with the major variant transitions.

From a public health perspective, these possible mechanisms have one important thing in common; they underscore the importance of widespread and equitable distribution of up-to-date vaccines, since saltational evolution in disadvantaged or remote populations carries a risk of emergence of new, highly transmissible variants.

While we have modelled each jump as a saltation occurring in a single individual, we should stress that we cannot rule out that the observed jumps occurred as a product of accelerated evolution in a chain of a few individuals, such as a string of immunocompromised individuals experiencing moderately increased pathogen mutation rates. The meager amount of data from outside Europe and North America (see S7B Fig) underscores that this cannot be ruled out, although substantially increased mutation rates would be required to account for the observed saltations. The plurality of potential etiologies highlights the need for comprehensive research into the mechanisms which may underlie the observed saltational/accelerated evolution. Such studies would be most welcome and would have to consider multiple scales, from molecular mechanisms and within-host evolution to the epidemiological dynamics which may contribute to saltations.

### Extensive sequencing is paramount

The type of analysis performed in this study requires large amounts of sequence data, beyond what could usually be obtained for infectious diseases prior to COVID-19. As shown in S2 Fig,

a similarly clear and detailed distribution of nucleotide distances could not be obtained for influenza H1N1 or H3N2. This is just one example of how incredibly useful the high level of genomic surveillance achieved for SARS-CoV-2 is, and more generally highlights the potential that extensive sequencing of pathogens holds for advancing phylodynamic understanding across pathogens [36]. While many countries have since scaled down the level of testing and sequencing of SARS-CoV-2, scientific insights based on this data will no doubt continue to emerge and have a lasting impact on our understanding of pandemics—as well as endemic infections—more broadly.

## The role of immunity

In our simulations so far, we have not explicitly modelled any effects of immune memory. We have allowed for new variants with higher effective reproduction numbers to arise, but have not distinguished between whether that advantage stemmed purely from higher infectiousness or from some degree of immune escape. However, it is worth noting that the empirical pattern of punctuated evolution held for every major transition up to and including Omicron BA.5 (Fig 1). When e.g. the Alpha variant became dominant, only about 3 infections per 100 people had been recorded in the United Kingdom [37] and vaccinations had not yet begun in earnest. While this is surely an undercount, a general depletion of susceptibles was not a main driver for the success or emergence of the Alpha variant. As such, the punctuated evolutionary pattern does not seem to be hinged on a connection between Hamming distance and evasion of immunity. In the case of the transition to Omicron, immune escape certainly played a role [38, 39], but it would seem that the mechanism of punctuated evolution is more general than that. In [40], the authors explicitly decompose fitness advantages into intrinsic and antigenic. Introducing a similar distinction in a genotypic fitness landscape model with saltation is an interesting possible extension of the present work.

## Exploring recent co-circulation of SARS-CoV-2 variants

As mentioned in the Results section, the sequence landscape has been increasingly complex since the transition to Omicron BA.5, with a higher degree of co-circulation and rapid strain turnover. In S3 Appendix, we extend our mathematical model with a simple implementation of (tunably) strain-specific immunity. Here, we find that incomplete cross-immunity between strains provides a selective pressure which can lead to co-circulation of several variants, even in the absence of intrinsic transmissibility advantages. Furthermore, the appearance of an intrinsically more transmissible variant into a heterogeneous immunity landscape does not necessarily lead to a diversity bottleneck. Rather, the development of the Hamming distribution depends on the levels of cross immunity (and, conversely, strain-specificity of immunity) between variants. Two possible outcomes are shown in Fig 7, with more details given in S3 Appendix. If cross immunity is absent (left panel), the appearance of a new highly transmissible variant is unlikely to lead to a homogenization of the antigenic landscape. However, if there is even partial cross-immunity between circulating variants (right panel), the emergence of a new intrinsically fitter variant is likely to 'refocus' the Hamming distribution and lead to a bottleneck. These simulations are highly conceptual in nature and by no means provide an exhaustive description of the late-2022 situation of SARS-CoV-2 co-circulation and rapid variant turnover, which we deem outside the scope of this paper. However, the simulations may provide some of the building blocks for such an analysis, which would be a highly worthwhile direction for future research.

 At the time of writing, a recombinant SARS-CoV-2 lineage by the name of XBB, including the particularly concerning member XBB.1.5, is circulating at appreciable levels in much of the

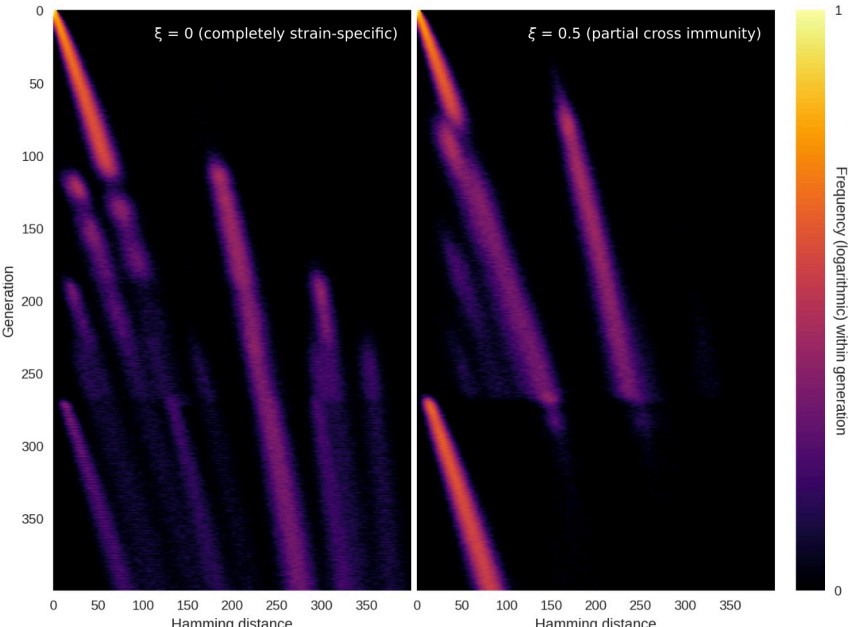

**Fig 7. Highly transmissible variant emerging in a heterogeneous immunity landscape.** In these simulations, we explore what happens when an intrinsically more transmissible variant emerges in a scenario with several co-circulating variants in a heterogeneous immunity background. See S3 Appendix for details on implementation, including the definition of the cross immunity parameter, $\xi$. **Left**: At $\xi = 0$, there is no cross immunity (e.g. immunity is completely strain-specific). In this case, co-circulation continues although a more transmissible variant is introduced at time $t = 250$. The new variant shows up as a peak at low Hamming distance, becoming visible around $t = 175$. **Right**: At $\xi = 0.5$, there is appreciable (albeit partial) cross immunity. In this case, the emergence of a new, more transmissible variant homogenizes the genomic landscape, with a single peak at low Hamming distance beginning to dominate around $t = 175$.

world [26] and is understood to have a transmissibility advantage [41]. It remains to be seen whether this variant will be 'homogenize' the Hamming distribution in the sense described above, by out-competing the several co-circulating variants.

Even in the presence of (partial) cross-immunity, there are good reasons to believe that saltations will continue to play a role in facilitating the emergence of new variants. As described by [42], accumulating immunity changes the fitness landscape of a pathogen over time, lowering some fitness peaks while rendering other peaks relatively more advantageous for the virus. Saltations can then enable the pathogen to reach those fitness peaks. Indeed, as we have seen, it is plausible that high levels of (more or less strain-specific) immunity in a population may increase the rate at which new strains emerge by saltation. Such a connection further underscores the importance of broadly effective and widely available vaccines as well as any measures which decrease the likelihood of accelerated evolution within hosts, with its risk of seeding saltation events.

## Future directions

As a consequence of the parsimony of our model, we have not explicitly modelled recombination events, but rather assume that each multi-site jump involves a random set of sites. Recombination has been reported in SARS-CoV-2, including—but not limited to—in conjunction with treatment of immunosuppressed patients [27, 28, 43, 44]. Future work could explore the implications of allowing for recombination events in this type of model. Doing so would

require a higher level of detail, resulting in a model that would conceivably be closer to biological 'ground truth' but not as parsimonious.

The influenza model of [17], which gives rise to Hamming dynamics reminiscent of the saltation-free simulations of Fig 4, does so in a very different way. There, it is assumed that the pathogen explores a *neutral network* (a set of antigenically and fitness-wise equivalent genotypes which are connected by one-mutation neighbours [18]) in the vicinity the prevailing strain. This goes on until the 'random walk' happens upon a configuration which is substantially antigenically different from the prevailing cluster, albeit connected to it by a single mutation. Once this happens, a new cluster emerges which has only limited cross-immunity with the prevailing strain. Since all steps along the way are small, the new variant will be very close (genotypically) to a member of the previous cluster. Consequently, this type of dynamics does not produce abrupt spikes in Hamming distance, such as the ones shown in Figs 1 and 3.

There are a few models in the literature that seek to address the connection between saltation, epistasis and the likelihood of emergence of new variants ([22] and [34], the latter of which is based on the model by [45]). However, in contrast to existing theoretical studies, we address the empirical temporal development of diversity and propose a model which can directly replicate the main features of that distribution.

We have focused on capturing the main features of the evolution of SARS-CoV-2 as parsimoniously as possible and although we have explored a number of biologically motivated extensions, our model still represents a theoretical foundation upon which more sophisticated models can be built. There is much to be done in terms of understanding and modelling the precise fitness landscape of SARS-CoV-2, including its dependence on host immunity history. More broadly, an increase in genomic surveillance across multiple pathogens will doubtlessly lead to new insights into the diversity dynamics of other pathogens. This would not only enable research into the evolution of individual pathogens, but allow us to question how co-circulating pathogens affect the diversity dynamics of one another.

## Materials and methods

### Temporally resolved Hamming distributions from sequence data

In this section, we describe the data processing workflow which was used to generate the Hamming distance plots of Figs 1 and 2. We have used the open, GenBank-derived dataset of aligned SARS-CoV-2 sequences from Nextstrain [2]. In our main analysis, we have used UK sequences from the 1st of March 2020 onwards. See also S1 Fig for an illustration of the following workflow:

- For each day $t$ in the interval:

  – Select 5,000 random pairs of whole-genome sequences (i.e. 10,000 sequences) obtained within a 1-week time window starting on day $t$.

  – For each pair of sequences $s_i$ and $s_j$:

    ∗ Go through both sequences, site by site, and record the number of differences between them, $H_{ij}$. This is the pairwise Hamming distance.

  – Compute a probability density/histogram $p_t(H)$ based on the observed Hamming distances $\{H_{ij}\}$.

It is then this function, $p_t(H)$, that is plotted in Fig 1A. In practice, we have used the metadata provided by Nexstrain, which contains fields describing the nucleotide differences relative to the reference strain Wuhan-Hu-1 (GenBank reference sequence accession number

MN908947.3), rather than operating directly on the whole-genome sequences. Numerically, this makes no difference, but it affords a large increase in performance, since it allows us to avoid processing unchanged regions of the genome, which do not contribute to the Hamming distance.

For Fig A in S2 Appendix, which instead shows the distance to the reference sequence (the 'absolute' Hamming distance), the above workflow is slightly altered:

- For each day $t$ in the interval:

  – Select 5,000 random whole-genome sequences obtained within a 1-week time window starting on day $t$.

  – For sequences $s_i$:

    * Go through the sequence, site by site, and record the number of differences $H_i$ between $s_i$ and the reference sequence. This is the absolute Hamming distance.

  – Compute a probability density/histogram $p_{0,t}(H)$ based on the observed Hamming distances $\{H_i\}$.

It is then this function, $p_{0,t}(H)$, that is plotted in Fig A in S2 Appendix.

## Branching model with saltational evolution

The mechanistic model developed for this study is a discrete-time branching model coupled to a genotypic fitness landscape model.

In the simulations of Figs 3 and 4, we assumed a constant prevalence, for simplicity. This amounts to keeping the mean effective reproductive number across the population at unity. In Fig 6 we relax this assumption and explore a version of the model with epidemic dynamics. We start by documenting the constant-prevalence version of the model, as well as the genotypic fitness landscape element, before we go on to describe how we incorporate SIRS dynamics and spatial structure.

**Evolutionary branching model with constant prevalence.**   In the model, each new generation of infections consists of a fixed number of individuals, $N$, and generations do not overlap. Consequently, there are $N$ infected individuals at any given time. Each infected individual $i$ has an associated bit-string $G_i$ of length $L$, representing the genome of the pathogen. We do not explicitly model any within-host diversity, as we are only interested in the genome of the pathogen that is eventually transferred during transmission.

At each time step (corresponding to one generation), a new random individual $i$ is repeatedly selected and allowed to infect a number $z_i$ of new individuals, selected from a Poisson distribution with mean $R_i$, i.e. $z_i \sim Pois(R_i)$. This continues until a total of $N$ new transmissions have occurred in that generation, ensuring that the prevalence is kept constant. At transmission, the pathogen genome of the infector is copied to the infectee. The personal reproductive number $R_i$ is determined by the fitness of the bit-string $G_i$, the details of which are discussed in the next subsection.

In each newly infected individual, there is a risk of mutation. The number of point mutations $m_i$ that occur within the $i$'th host is drawn from a distribution. In the case of homogeneous mutation (i.e. absence of saltation), $m_i$ is drawn from a Poisson distribution characterized by a mutation rate $\mu_0 < 1$. Saltation, on the other hand, is simulated by drawing $m_i$ from a bimodal distribution characterized by two different mutation rates/sizes $\mu_0$ and $\mu_1$, ensuring that an outsized amount of mutation can take place within a single host on rare

occasions. Concretely, we have used the distribution $P_s(m)$ given by:

$$P_s(m) = (1 - \epsilon)\text{Pois}(m; \mu_0) + \epsilon U(m; \mu_1 \pm \Delta\mu) \tag{1}$$

Where $U(m; \mu_1 \pm \Delta\mu])$ is the uniform distribution centered on $\mu_1$ with half-width $\Delta\mu$, and Pois $(m; \mu_0)$ is a Poisson distribution with mean $\mu_0$. $\epsilon \ll 1$ is a small dimensionless quantity measuring the frequency of saltational mutation. The parameter $\mu_0$ gives the rate of non-saltational mutation while $\mu_1$ is the typical size of a saltation.

We use this simple bimodal distribution out of convenience, but our results do not change qualitative if another bimodal distribution is used.

Once the quantity $m_i$ has been drawn, a number $m_i$ of random bit flips are then performed in the genomic bitstring $G_i$, each flip corresponding to a point mutation.

## Modelling sign epistasis

Before simulations start, a number of $N_e$ of 'epitopes' (regions in the genome on which fitness depends), each of length $L_e$, are designated. We assume non-overlapping epitopal regions and thus require $L_e N_e \leq L$.

Within each epitope, a number $N_H$ of highly fit combinations are assigned. We have assumed $N_H = 1$ for all of our simulations, but since the general $N_H$ case is no more complicated, we include the parameter here. The fitness of each combination is measured in terms of its contribution $\delta R_H$ to the individual reproductive number. In general, $\delta R_H$ for each combination may be drawn from a distribution $P_H(\delta R_H)$ to allow for a variety of combinations with different fitness values.

Tunable sign epistasis is modeled by assigning a fitness contribution $\delta R_L \leq 0$ to each combination which lies within a Hamming distance $d_0$ of a high-fitness combination. The overall fitness of a given genotype is then obtained by adding up the contributions for each of the $N_e$ epitopes:

$$R_0 = R_0^{\text{initial}} + \sum_{i=1}^{N_e} \delta R_i, \tag{2}$$

with the constraint that $R_0 \geq 0$. In practice this constraint is enforced by letting

$$R_0 = \max\left(0, R_0^{\text{initial}} + \sum_{i=1}^{N_e} \delta R_i\right), \tag{3}$$

High sign epistasis is then achieved when $d_0 > 1$ and $\delta R_L \ll 0$. However, the model also allows for incomplete or partial sign epistasis: if $\delta R_L$ for each combination is drawn from a distribution $P_L(\delta R_L)$ which has support at $\delta R_L = 0$, then each peak in the fitness landscape will not be completely surrounded by troughs. In other words, in that case it may be possible to evolve to a highly fit variant through a series of single point mutations without suffering decreased fitness in the process.

Unless otherwise specified, we run our simulations with the parameter values given in Table 1.

## Incorporating SIRS dynamics

In the simulations of Figs 3 and 4 we assumed constant incidence, meaning that the number of infected within any given generation was $I(t) = I_0$ with $I_0$ a constant (thus, prevalence was constant as well). However, to relax this assumption we incorporate susceptible-infected-recovered-susceptible (SIRS) dynamics.

**Table 1. Model parameters and their values.**

| Parameter | Description | Value (base case) |
|---|---|---|
| $N$ (constant-prevalence simulations) | Infected population size | 50,000 (= prevalence) |
| $N$ (agent-based SIRS simulations) | Total population size | $2 \times 10^6$ (= $S + I + R$) |
| $L$ | Genome length (bits) | 1000 |
| $L_e$ | Length of each epitopal sequence | 5 |
| $N_H$ | Number of highly fit configurations of each epitope | 1 |
| $N_e$ | Number of epitopes | 5 |
| $d_0$ | Width of troughs in fitness landscape | 3 |
| $\langle \delta R_H \rangle$ | Avg. change in $R_0$ due to beneficial genotype | 1 |
| $\langle \delta R_L \rangle$ | Avg. change in $R_0$ due to deleterious genotype | $-\infty$ (no transmission) |
| $\mu_0$ | Base mutation rate (for whole genome) | 0.3 |
| $\epsilon$ | Frequency of saltation | 0 or 0.0001 |
| $\mu_1$ | Typical size of saltations | 150 |
| $\Delta\mu$ | Half-width of saltation size distribution | 50 |
| $T_{ij}$ | Relative transmission rate betw. populations $i$, $j$. | 0–1 |
| $\omega$ | Rate of waning of immunity in SIRS simulations | 0.04/generation |

In order to achieve this (and to simplify the later addition of strain-specific immunity to the model), we implement a discrete-time agent-based version of our model, in which we also track susceptible and recovered individuals, and not just the infected population. We denote the total number of susceptible, infected and recovered individuals in generation $t$ by $S(t)$, $I(t)$ and $R(t)$, respectively. Here, we detail the version of the dynamics with complete cross-immunity (i.e. recovered individuals are immune to all variants until immunity wanes). A version with strain-specific immunity is discussed in S3 Appendix. The simulations proceed as follows. At time $t = 0$, let:

$$
\begin{aligned}
S(t = 0) &= N - I_0, \\
I(t = 0) &= I_0, \\
R(t = 0) &= 0.
\end{aligned}
$$

Each infected person will cause a number of new infection determined by their effective reproductive number, which is given by the basic reproductive number of the strain they are infected with, discounted by the current fraction of susceptible individuals to model susceptible depletion. Each recovered person has a constant probability rate $\omega$ for becoming susceptible once again. In other words, this is modeled as a Poisson process with rate $\omega$. Note that this not only corresponds to waning of immunity, but also to any other mechanism by which a recovered individual may become replaced by a susceptible one (such as population turnover). However, we will refer to $\omega$ as the rate of waning. In our simulations (Fig 6 and S6 Fig), we set $1/\omega = 25$ meaning that duration of immunity averages 25 generations. This figure is not supposed to reflect any particular value for SARS-CoV-2, but is rather used to illustrate the robustness of the pattern of punctuated evolution to waning immunity. In the interest of simplicity, we have ignored any seasonal effects on transmission. We consider this a reasonable simplification, both due to the conceptual nature of our model and the understanding that susceptible dynamics rather than seasonality is the major limiting factor in the pandemic phase [46].

## Spatial structure

We implement a minimal model of spatial structure by incorporating a metapopulation element. Let there be $n_{\mathrm{pops}}$ populations, each with total population $N_i$ ($i \in \{1, \ldots, n_{\mathrm{pops}}\}$). At time $t = 0$, let the number of susceptible, infected and recovered individuals in each population be given by:

$$
\begin{aligned}
S_i(t = 0) &= N_i - I_{i,0}, \\
I_i(t = 0) &= I_{i,0}, \\
R_i(t = 0) &= 0.
\end{aligned}
$$

In our simulations (S6 Fig) we assume identical population sizes, $N_i = N/n_{\mathrm{pops}}$, and an initial equipartitioning of infected individuals $I_{i,0} = I_i/n_{\mathrm{pops}}$, where $N = \Sigma_i N_i$ and $I_0 = \Sigma_i I_{i,0}$.

The transmission rate between populations is then determined by the matrix elements $\beta_{ij} = \beta T_{ij}$ where each element $T_{ij}$ gives the relative transmission rate from population $i$ to $j$ and $\beta$ represents the transmissibility of the strain the infected individual carries. We assume that **T** is a symmetric matrix, $T_{ij} = T_{ji}$.

In S6 Fig we took **T** to have the following form:

$$
T = \begin{bmatrix} 1 - \varepsilon & \varepsilon & 0 \\ \varepsilon & 1 - 2\varepsilon & \varepsilon \\ 0 & \varepsilon & 1 - \varepsilon \end{bmatrix} \tag{4}
$$

with $\varepsilon$ at either 0 (panel A), $10^{-4}$ (panel B) or $10^{-3}$ (panel C). This corresponds to a linear layout of three populations, with transmission occurring only between adjacent compartments.

## Modelling decreasing absolute Hamming distance

As described in S2 Appendix, the typical Hamming distance between circulating genomes and the ancestral variant is not necessarily monotonically increasing with time. We call this distance the *absolute* Hamming distance, in contrast to the pairwise distance between concurrently circulating genomes which we call the *relative* Hamming distance (to reflect that the absolute Hamming distance is measured with respect to a fixed point in the genomic space).

We begin by describing a very simple variation upon the model which has the effect of allowing the absolute Hamming distance to decrease (as well as increase) at variant transitions. In this section, we assume a constant prevalence of $N$ infected individuals.

Assume that a fraction $p_d$ of the first generation (i.e. $p_d N$ individuals) have prolonged infections, lasting $\tau_d$ typical generations before onward transmission. Assume furthermore that mutations happen at a rate $\mu_d$ for these individuals, such that a number of point mutations, $\mu_d \tau_d$, occurs before onward transmission. Here, $\mu_d$ is the mutation rate associated with these prolonged infections. The rest of the population is assumed to be homogeneous with respect to the occurrence of mutations, all possessing a mutation rate $\mu_0$. We draw $\tau_d$ from a uniform distribution with support throughout the entire simulation (which is assumed to have duration $t_f$), $\tau_d \sim U(\tau_d; t_f/2 \pm t_f/2)$. Furthermore, the fitness advantage $\delta R_H$ of different epitope configurations was drawn from a uniform distribution as well, to avoid fitness degeneracy (multiple equally fit variants).

This simple modification of the model enables a non-monotonic time development of the absolute Hamming distance, as shown in Fig B in S2 Appendix (panel B), while preserving the dynamics of relative Hamming distance shown in Fig 3.

This is of course a highly simplistic variation upon the base model, but it serves to show that prolonged infection or introduction of (mutated versions of) previous variants can account for absolute Hamming distance sometimes decreasing at variant transitions.

## Supporting information

**S1 Fig. Data analysis workflow.** To generate the Hamming distribution for a given point in time, all sequences sampled within a week-long window starting on the given day are pooled. Then, pairs of sequences are repeatedly selected at random from this sequence pool, and the pairwise Hamming distance (number of sites which differ) is computed. All the computed Hamming distances are then pooled and a distribution (histogram) is generated.
(TIF)

**S2 Fig. Hamming distributions for influenza H3N2 and H1N1.** Based on the Hemagglutinin (HA) gene. With influenza, the amount of genomic surveillance data is much more limited and the temporal Hamming distributions are less well-defined. In order to ensure sufficient data for each time point, a sampling window of 30 days was used, as opposed to the 7 days used for SARS-CoV-2 in the main text.
(TIF)

**S3 Fig. Model schematics. A)** The fitness landscape and epistasis components of the model. The majority of the fitness landscape is assumed neutral. In the case of gradual evolution devoid of saltation (top), the pathogen performs a random walk in this neutral space until it hits upon a deleterious configuration. As a model of sign epistasis, beneficial configurations are surrounded by deleterious ones. In the case of gradual evolution, the deleterious regions are unlikely to be traversed before the lineage dies out. However, in the case of saltational evolution (bottom), several point mutations may occasionally happen in the same genome within the same generation, leading to a jump which can enable the pathogen to bypass a deleterious region. Note that this is only a 1-dimensional conceptual representation of a highly multidimensional fitness landscape. **B)** In each generation of the branching model, each individual stochastically infects $z$ new individuals. Upon transmission, the pathogen genome (depicted as a string of black and white squares) is inherited. Occasionally a point mutation will occur, as indicated in the lower right genome. In the case of saltation (see panel A), multiple such point mutations can occur within the same genome in the same generation.
(TIF)

**S4 Fig. Temporary coexistence of two equally fit variants.**
(TIF)

**S5 Fig. Saltational evolution in the absence of sign epistasis.** When saltational evolution is allowed, but epistasis is absent or very weak, a mixture of qualitatively different transitions occur. Some resemble the diversity spikes seen in Fig 3, but more commonly transitions will involve a gradual, linear increase in diversity followed by a collapse, as seen in Fig 4. **A)** Time evolution of the Hamming distance distribution. For each generation indicated on the vertical axis, the colour encodes the histogram of Hamming distances between genomes within that generation. **B)** Time evolution of the mean and median Hamming distance between genomes present in any given generation of the model simulation. In these simulations, $\delta R_L = 0$ (no epistasis) while saltations were of typical size $\mu_1 = 150$.
(TIF)

**S6 Fig. Simulations with spatial (metapopulation) structure.** Here we simulate the same SIRS dynamics as in Fig 6, but in a metapopulation consisting of multiple subpopulations. **A)**

Here we probe the significance of the level of transmission *between* populations. The within-population transmission rate $T_{ii} \approx 1$ ($i \in \{1, 2, 3\}$) is assumed much greater than the between-population transmission rate $T_{ij}$ (with $j = i \pm 1$). **(Left)** With inter-population transmission rate $\beta_{i,i+1} = 0$, mutations never spread from one population to another and coexistence of variants with different fitness can last indefinitely. **(Middle)** With an inter-population transmission rate of $10^{-4}$, transitions are severely prolonged but coexistence of variants with different fitness values does not last indefinitely. **(Right)** At an inter-population transmission rate of $10^{-3}$, transitions are only moderately prolonged compared to the non-spatial dynamics of Fig 6. **B)**.
(TIF)

**S7 Fig. Hamming distributions based on global sequences. A)** Hamming distribution based on available SARS-CoV-2 sequences, regardless of origin. **B)** Hamming distribuion computed on the basis of sequences from outside of Europe and North America. These comprise approximately 1.4% of the global sequences (i.e. of those included in panel A).
(TIF)

**S1 Video. Animated Hamming distribution.** Day-by-day time development of the Hamming distribution for UK samples obtained between March 2020 and November 2022. Each snapshot is based on samples obtained within a one-week time window. The insert shows the fraction of UK sampled sequences belonging to each variant. The EU1 (B.1.177) cluster, which preceded the Alpha variant in the UK, is shown as well.
(MP4)

**S1 Appendix. Diversity dynamics based on US sequences.**
(PDF)

**S2 Appendix. Fitting the origin-centered Hamming distribution.**
(PDF)

**S3 Appendix. Variant dynamics under strain-specific immunity.**
(PDF)

## Acknowledgments

We would like to thank the members of the Grenfell, Levin and Metcalf Labs at The Department of Ecology and Evolutionary Biology, Princeton University, for fertile plenary discussions, with special thanks to Daniel Park, Qiqi Yang, Luojun Yang, Inga Holmdahl, Nicole Nova and Justin Sheen. We would also like to thank Arne Traulsen for enlightening discussions pertaining to the formulation of our model and Christian Berrig and Viggo Andreasen at the PandemiX Center, Roskilde University, for much appreciated comments on data visualization.

## Author Contributions

**Conceptualization:** Bjarke Frost Nielsen, Lone Simonsen, Simon A. Levin, Bryan T. Grenfell.

**Data curation:** Bjarke Frost Nielsen.

**Formal analysis:** Bjarke Frost Nielsen, Chadi M. Saad-Roy, Yimei Li, Kim Sneppen, Cécile Viboud, Simon A. Levin, Bryan T. Grenfell.

**Funding acquisition:** Kim Sneppen, Lone Simonsen, Simon A. Levin, Bryan T. Grenfell.

**Investigation:** Bjarke Frost Nielsen, Chadi M. Saad-Roy, Yimei Li, Kim Sneppen, Lone Simonsen, Cécile Viboud, Bryan T. Grenfell.

**Methodology:** Bjarke Frost Nielsen, Chadi M. Saad-Roy, Kim Sneppen, Lone Simonsen, Cécile Viboud, Simon A. Levin, Bryan T. Grenfell.

**Software:** Bjarke Frost Nielsen.

**Supervision:** Kim Sneppen, Lone Simonsen, Cécile Viboud, Simon A. Levin, Bryan T. Grenfell.

**Validation:** Bjarke Frost Nielsen, Chadi M. Saad-Roy, Kim Sneppen, Cécile Viboud, Bryan T. Grenfell.

**Visualization:** Bjarke Frost Nielsen, Chadi M. Saad-Roy, Yimei Li, Kim Sneppen, Bryan T. Grenfell.

**Writing – original draft:** Bjarke Frost Nielsen, Yimei Li, Kim Sneppen, Lone Simonsen, Cécile Viboud, Simon A. Levin, Bryan T. Grenfell.

**Writing – review & editing:** Bjarke Frost Nielsen, Chadi M. Saad-Roy, Yimei Li, Kim Sneppen, Lone Simonsen, Cécile Viboud, Simon A. Levin, Bryan T. Grenfell.

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
