## [Decision Letter · Decision Letter 0]

28 Nov 2022

Dear Dr. Nielsen,

Thank you very much for submitting your manuscript "Host heterogeneity and epistasis explain punctuated evolution of SARS-CoV-2" for consideration at PLOS Computational Biology.

As with all papers reviewed by the journal, your manuscript was reviewed by members of the editorial board and by several independent reviewers. In light of the reviews (below this email), we would like to invite the resubmission of a significantly-revised version that takes into account the reviewers' comments.

We cannot make any decision about publication until we have seen the revised manuscript and your response to the reviewers' comments. Your revised manuscript is also likely to be sent to reviewers for further evaluation.

Sincerely,

Alexandre V. Morozov, Ph.D.

Academic Editor

PLOS Computational Biology

Thomas Leitner

Section Editor

PLOS Computational Biology

Reviewer's Responses to Questions

**Comments to the Authors:**

Reviewer #1: Summary: This paper analyzed SARS-CoV-2 sequence data using Hamming distance to find good evidence for evolution of this virus primarily through saltation with epistasis. They validate their finding using simulations. Overall, I am impressed with the quality of this paper. It is a very useful contribution to the understanding the evolution of this virus. I have a few questions and concerns listed below. I hope the authors will address these points and incorporate them into the paper.

1. The authors claim that most of the evolution of SARS-CoV-2 happens in immunocompromised individuals in whose immune landscape the virus has greater freedom to evolve. Although this makes sense, I would like to see a discussion of other possible mechanisms that might create the kinds of saltation events observed. These may happen gradually in a much larger fitness landscape than what is captured by databases consisting of sequences from symptomatic individuals. In the discussion section the authors consider “reverse zoonosis” to create a mixed epistatic landscape connecting humans to animals. A possible discussion of other mechanisms that might also create a larger epistatic landscape (list given below) could perhaps also be included.

• Changes in binding location from lung, lower respiratory tract to upper respiratory tract to make the virus more infective and less virulent (there is evidence that this is happening in the variants which have emerged since March-April 2020). This adaptation might create sufficient drift followed by a few mutations for a new variant to appear suddenly and be seen as saltation . Is this a possibility?

• Effect of asymptomatic infected individuals in whom the virus may evolve without being sequenced and create novel variants. The pool of asymptomatic infected individuals (the “dark matter” of this pandemic) may be very large compared to the identified, symptomatic cases, particularly as the virus becomes more infective and less virulent.

• Changes induced by vaccination targeted to specific variants may create favorable landscapes for new variants.

• Changes in behavior of individuals, relaxation of quarantine, mask requirements, increased travel, more time spent in areas of poor vaccination (aircraft) or with larger groups of people as controls are relaxed might also create new pools of infected individuals who might potentially provide new variants.

• Opening of schools, colleges, and businesses.

2. The authors analyze mostly data from first would countries (UK, USA). However, one would expect much greater evolution of disease in third world or densely populated countries with poor health services, such as Africa, India, the Middle East. Would it be possible to include and analyze at least a few sequences from such countries?

3. Is there sequence data from random sampling of the UK population to identify asymptomatic carriers? If so, it would be interesting to check whether these individuals have non-synonymous mutations that might bridge the gap between saltation events.

4. On page 6, referring to within patient evolution in immunocompromised individuals the authors state that “the most well-documented is perhaps elevated mutation in immunocompromised individuals.” This is an important point and needs a reference.

The authors also describe in detail the simulations they did to test and validate their conclusions. Unfortunately, I am not an expert on such simulations so I cannot comment on them. I presume other, more competent reviewers can address this.

This is a well written and timely paper. The point it makes about universal and equitable distribution of vaccines worldwide is an important conclusion.

Reviewer #2: The authors provide results of several important types. First, they perform a detailed analysis of genomic data from Covid-19 infections in the UK over an extended period, and calculate the Hamming distance between genomes in the set, determining the genomic diversity over time. They also do this for influenza data, although the data available is not as extensive and high quality. They note that the Covid-19 data appears to have a punctuated evolution, with genetic drift alternating with large increases of diversity followed by a crash. Influenza does not seem to show the same sharp peaks, and the authors seek to model this defining feature of the Covid-19 data. They posit that it is due to genetic saltation, rare but large jumps in critical genetic sequences that enable the virus to traverse deep “valleys” in a fitness landscape. They model this with a commendably simple but effective model, and also add SEIR modeling to include the effects of different populations. In all cases they find they need to have saltational evolution in order to get the qualitative features of the data.

The paper is very clearly written in understandable colloquial English, the figures are all of them excellent and clear (although not clear without viewing the figure caption, since there is precious little identifying information on each figure. However, since in any publication setting, the figures will appear with their captions, I do not list this as something to be amended.) The Supplements are also commendably short, and yet clearly written and on topics of relevance to the main paper. The supplementary figures are likewise very useful.

There are only a very few points that need addressing.

1) The word “important” is repeated twice in a row, the only typo I found. This is on page 3, the paragraph two above the Results section. Search for “important important” in the paper and it will be found.

2) (a) This next point is much more pertinent, and should be addressed in the manuscript. Namely, the time dependence of genetic diversity can suggest another effect, independent of any saltational mutations: the punctuated nature of people's interactions in this pandemic, very different dynamics than happens with influenza. A sudden law to shut down businesses, a sudden law to mandate masks, a closing of schools, then in reverse, the masks off, then back on a few months later, schools open but with masks, then suddenly masks off, then suddenly people gathering, then larger groups, etc.

In other words, could the jumps in genetic diversity be due not to saltational mutation, but rather if everyone’s mutation was fairly slow, but people being separated, the mutation was for extended periods in many different directions, then suddenly people appeared out with their new mutations and infected others? Many, many people got Covid who did not necessarily go to the doctor and get sequenced, especially with the later milder versions. It could seem that this “punctuated appearance” of people and groups could also just as easily explain the sudden jumps. Superspreader events, for example, especially if some low-immunity individuals attended (say with a higher rate of mutation but not a saltational one).

(b) In a second related question, how can a presumably constant (low) rate of saltational viral evolution explain the accelerated appearance of closely related new strains of the virus? This is data not covered in the manuscript, more recent, but surely the authors are aware of it. Many, many new variants, not the well-separated peaks of the author’s results. But could be explained (?) by more and more people finally coming out without masks, everyone with their own new variant.

(c)The interactions between groups were attempted to be addressed with the authors’ SEIR modeling, but those are smooth differential equations, and do not have the punctuated nature of the changes. Could the authors comment on what could happen in their model if they did not have saltational evolution, but rather the sudden appearance (in public) of people with different strains? Could that explain the punctuated rises and falls of genetic diversity?

Points a, b, and c above do not need a large change to the paper: just a sentence or so, say in the results or conclusions, to say whether this is an alternative effect that should be considered or if not, why not.

The introduction does an excellent job of saying that saltation might be the way to model the data, not that it must.

3) Figure 5 is puzzling somewhat: without saltation, the viruses in this model all die out. This is not what happens with most viruses; they persist in individuals and mutate, and come back to infect next season or next time someone has lowered immunity. The flu is here year after year, and certainly with a reproductive rate greater than one, and the authors do a good job of explaining how the model would be altered to model the flu well. Hepatitis is another one that persists, at high levels in some communities, without any saltation. Figure 4 is fine; what collapses after the genetic drift is the genetic diversity, not necessarily the viral population (which still presumably has a quasispecies distribution of some width). But Figure 5 seems to say that in its non-saltation version this model results in a non-functional virus. Perhaps the situation in Figure 5 can be clarified.

4) The branching part of the model could be explained with more clarity, in relation to Figure S3B. The size of the population N is not in Table I. The text describes the process fairly well, and Figure S3A is very good, but it is not clear how the images of Figure S3B relate to the description.

**Have the authors made all data and (if applicable) computational code underlying the findings in their manuscript fully available?**

Reviewer #1: Yes

Reviewer #2: Yes

PLOS authors have the option to publish the peer review history of their article (what does this mean?). If published, this will include your full peer review and any attached files.

Reviewer #1: No

Reviewer #2: No
---

## [Decision Letter · Decision Letter 1]

25 Jan 2023

Dear Dr. Nielsen,

We are pleased to inform you that your manuscript 'Host heterogeneity and epistasis explain punctuated evolution of SARS-CoV-2' has been provisionally accepted for publication in PLOS Computational Biology.

Best regards,

Alexandre V. Morozov, Ph.D.

Academic Editor

PLOS Computational Biology

Thomas Leitner

Section Editor

PLOS Computational Biology

Reviewer's Responses to Questions

**Comments to the Authors:**

Reviewer #1: Good response to reviewers !

Reviewer #2: I thank the authors for their clearly substantial efforts to improve their manuscript. I now conclude the authors have addressed all issues, as far as I can tell. Moreover, in the new version are added additional very interesting information. The new information not only included additional data, but also additional types of analysis based on referee comments, and this analysis improves the paper very much. In effect, the analysis disproves all possible alternative reasons for a saltation in viral quasispecies composition, showing that suggested alternative scenarios give a gradual blending until the more fit virus wins, and not saltation. I now believe this paper contains exciting new results, showing 1) saltation exists in this virus 2) the data to prove it, and 3) how to best analyze the data, as well as how to add additional test scenarios to their modeling. The figures have been made much more clear, and the new Supplement is very good. No further issues to note.

**Have the authors made all data and (if applicable) computational code underlying the findings in their manuscript fully available?**

Reviewer #1: Yes

Reviewer #2: Yes

PLOS authors have the option to publish the peer review history of their article (what does this mean?). If published, this will include your full peer review and any attached files.

Reviewer #1: No

Reviewer #2: No

---

## [Editor Report · Acceptance letter]

7 Feb 2023

PCOMPBIOL-D-22-01235R1 

Host heterogeneity and epistasis explain punctuated evolution of SARS-CoV-2

Dear Dr Nielsen,

I am pleased to inform you that your manuscript has been formally accepted for publication in PLOS Computational Biology. Your manuscript is now with our production department and you will be notified of the publication date in due course.

With kind regards,

Zsofia Freund
